Phylogeographic structure and northward range expansion in the barnacle Chthamalus fragilis

Govindarajan Annette F. 1 afrese@whoi.edu
Bukša Filip 1 2
Bockrath Katherine 3
Wares John P. 3 4
Pineda Jesús 1
1 Biology Department, Woods Hole Oceanographic Institution , Woods Hole, MA , USA
2 Department of Biology, University of Zagreb , Croatia
3 Department of Genetics, University of Georgia , Athens, GA , USA
4 Odum School of Ecology, University of Georgia , Athens, GA , USA
Toonen Robert
Electronic publication date: 2015 Apr 30
Publication date: 2015
Volume: 3
Electronic Location ID: e926
Received 2015 Mar 18; Accepted 2015 Apr 11
Copyright: © 2015 Govindarajan et al.
Copyright year: 2015
Copyright holder: Govindarajan et al.
License: This is an open access article distributed under the terms of the Creative Commons Attribution License, which permits unrestricted use, distribution, reproduction and adaptation in any medium and for any purpose provided that it is properly attributed. For attribution, the original author(s), title, publication source (PeerJ) and either DOI or URL of the article must be cited.
License URL: https://creativecommons.org/licenses/by/4.0/

Keywords: Chthamalus, Range expansion, Barnacle, Phylogeography

Funding: Woods Hole Oceanographic Institution NSF Biological Oceanography #1029526 Funding was provided by the Woods Hole Oceanographic Institution in an Independent Study Award to J Pineda and AF Govindarajan, a WHOI summer fellowship to F Bukša, and NSF Biological Oceanography #1029526 to JP Wares. The funders had no role in study design, data collection and analysis, decision to publish, or preparation of the manuscript.

==============================
The barnacle Chthamalus fragilis is found along the US Atlantic seaboard historically from the Chesapeake Bay southward, and in the Gulf of Mexico. It appeared in New England circa 1900 coincident with warming temperatures, and is now a conspicuous member of rocky intertidal communities extending through the northern shore of Cape Cod, Massachusetts. The origin of northern C. fragilis is debated. It may have spread to New England from the northern end of its historic range through larval transport by ocean currents, possibly mediated by the construction of piers, marinas, and other anthropogenic structures that provided new hard substrate habitat. Alternatively, it may have been introduced by fouling on ships originating farther south in its historic distribution. Here we examine mitochondrial cytochrome c oxidase I sequence diversity and the distribution of mitochondrial haplotypes of C. fragilis from 11 localities ranging from Cape Cod, to Tampa Bay, Florida. We found significant genetic structure between northern and southern populations. Phylogenetic analysis revealed three well-supported reciprocally monophyletic haplogroups, including one haplogroup that is restricted to New England and Virginia populations. While the distances between clades do not suggest cryptic speciation, selection and dispersal barriers may be driving the observed structure. Our data are consistent with an expansion of C. fragilis from the northern end of its mid-19th century range into Massachusetts.

Introduction

Evaluation of population genetic discontinuities and range boundaries in coastal marine species is essential for understanding the consequences of anthropogenic stressors like climate change which may be driving range shifts, particularly poleward range expansions (e.g., Barry et al., 1995; Zacherl, Gaines & Lonhart, 2003; Dawson et al., 2010; Harley, 2011). Along the Atlantic coast of the US, Cape Hatteras and Cape Cod are especially important boundary regions (Pappalardo et al., 2014). However, because these boundaries are permeable (e.g., many species traverse the boundaries; Pappalardo et al., 2014), as are other coastal boundary regions for nearshore species (e.g., Valentine, 1966), it is necessary to evaluate each species individually. The intertidal barnacle Chthamalus fragilis is currently found along the eastern United States, extending from the Gulf of Mexico to the Atlantic coast northward up to Massachusetts (Wells, 1966; Zullo, 1963; Carlton, Newman & Pitombo, 2011), and is thought to be experiencing a northward range expansion linked to warmer temperatures (Wethey, 1984; Carlton, Newman & Pitombo, 2011). Prior to the late 19th century, C. fragilis was observed from the Chesapeake Bay area and southward. It was first observed in New England (Woods Hole, Massachusetts) in 1898, and subsequently was observed in other locations south of Cape Cod, in Buzzards Bay and Vineyard Sound (Carlton, Newman & Pitombo, 2011). More recently, it is found along the north shore of Cape Cod, from the outer Cape (Provincetown) to Sandwich at the northern end of the Cape Cod Canal (Zullo, 1963; Carlton, 2002; Wethey, 2002; Jones, Southward & Wethey, 2012). C. fragilis is a conspicuous species occupying the easily accessible upper intertidal, so it is unlikely that an earlier northern presence was overlooked, particularly as the Woods Hole region has a long history of faunal surveys.

The source of the northern C. fragilis populations is controversial. It is unknown if the barnacles dispersed via natural (e.g., ocean currents) or anthropogenic vectors (e.g., ship hull fouling), or both. C. fragilis possesses a typical biphasic life cycle, with the potential for long distance dispersal. Adults are hermaphroditic with internal fertilization and are capable of self-fertilization (Barnes & Barnes, 1958). Thus, clusters of adults are not required for reproduction as in many barnacles (Crisp, 1950). Larvae are released into the water, typically in the summer (Lang & Ackenhusen-Johns, 1981), where they pass through 6 naupliar stages and a non-feeding cyprid stage. In chthamalids, the planktonic period may last up to three weeks or more (Miller et al., 1989), allowing ample time for larval transport by ocean currents. Cyprids settle on hard intertidal substrata and metamorphose into the adult form.

C fragilis settles on artificial surfaces, and thus has a high potential for dispersal by anthropogenic transport. Sumner (1909) suggested that the relatively sudden appearance of C. fragilis in Woods Hole, MA was due to human introduction. In support of this hypothesis, Carlton, Newman & Pitombo (2011) points out that Woods Hole was home to the Pacific Guano Company between 1863 and 1889, which received potentially fouled ships from South Carolina, the type locality for C. fragilis, and elsewhere. The construction of structures such as docks, pilings, and seawalls may have provided suitable habitats along the mostly sandy shoreline south of Connecticut, also facilitating range expansion (e.g., Jones, Southward & Wethey, 2012).

The New England region has experienced warmer temperatures since the 1850s (Carlton, 2002), and warmer temperatures may have facilitated the successful dispersal and establishment of C. fragilis by releasing it from competition with the less heat-tolerant barnacle Semibalanus balanoides in the upper intertidal (Wethey, 2002). In these intertidal areas, C. fragilis is found higher, where S. balanoides, the better competitor, cannot survive (Wethey, 2002).

The goals of this study were to investigate the phylogeographic structure of C. fragilis and gain insight into the origin of northern C. fragilis populations by comparing mitochondrial cytochrome c oxidase (COI) haplotypes from several locations in Massachusetts and Rhode Island with those obtained from locations farther south, in Virginia, South Carolina, Georgia, and Florida. Thus, sampling covered a ∼2,000 km range (minimum linear separation). While confirming the source of populations that are cryptogenic (i.e., of unknown origin) can be difficult, the existence of private haplotypes shared between the northern populations and a subset of southern populations may indicate the colonization pathway (Geller, Darling & Carlton, 2010). For example, private haplotypes shared between northern and South Carolina barnacles may support the idea that barnacles arrived through transport associated with the Woods Hole guano industry (Carlton, Newman & Pitombo, 2011). Alternatively, private haplotypes shared only between northern and Chesapeake Bay—area barnacles (at the northern end of their historic range) may suggest a range expansion. We compare genetic diversity and the distribution of mitochondrial haplotypes from barnacles ranging from Massachusetts to Florida, and demonstrate significant genetic structuring between northern and southern populations. We discuss the implications of these patterns for a genetic break near Cape Hatteras and the origin of northern C. fragilis.

Materials & Methods

We collected 108 Chthamalus fragilis individuals from 11 sites along the Atlantic and Gulf coasts of North America (Table 1). We extracted genomic DNA using DNEasy Blood and Tissue and Puregene kits (Qiagen) and amplified the mitochondrial cytochrome c oxidase I (COI) gene using standard primers (Folmer et al., 1994) and protocols. We ran 25 µl PCR reactions containing 1 µl of genomic DNA in a PCR program consisting of an initial denaturation at 95°for 3 min; 35 cycles of 95° for 30 s, 48° for 30 s, and 72° for 1 min; and a final extension at 72° for 5 min. We visualized PCR products on a 1.5% agarose gel stained with GelRed (Biotium). PCR products were purified using Qiaquick PCR Purification kits (Qiagen, Hilden, Germany) and quantified using a Nanodrop 2000 spectrophotometer (Nanodrop Technologies, Wilmington, Delaware, USA). Purified products were sent to MWG Eurofins Operon for sequencing in both directions.

Table 1 Summary statistics calculated in DnaSP.

None of the Tajima’s D values were significant.

Collection site	n	H	Hd	Pi	Tajima’s D	Fu’s Fs	
SAND	11	11	1	0.01640	−0.95411	−4.172	
TR	9	9	1	0.01749	−0.30109	−2.642	
WH	12	11	0.985	0.01750	−0.56516	−2.648	
RWU	11	8	0.945	0.01222	−0.01230	−0.431	
VA	11	10	0.982	0.01886	−0.27640	−1.849	
SC	12	11	0.985	0.01508	−0.31326	−3.116	
SI	7	7	1	0.01725	−0.23160	−1.463	
SA	7	7	1	0.01041	−1.43806	−2.449	
SK	7	7	1	0.01717	−0.63213	−1.471	
KL	14	14	1	0.01750	−0.64068	−6.227	
TF	7	6	0.952	0.00792	−0.46339	−1.188	
Total	108	93	0.996	0.01789	−1.70904	−109.686	
Notes.

N number of individuals

H number of haplotypes

Hd haplotype diversity

Pi nucleotide diversity

SAND Sandwich, Massachusetts

TR Truro, Massachusetts

WH Woods Hole, Massachusetts

RWU Bristol, Rhode Island

VA Gloucester Point, Virginia

SC Charleston, South Carolina

SI Sapelo Island, Georgia

SA Savannah, Georgia

SK Summerland Key, Florida

KL Key Largo, Florida

TF Tampa, Florida

We assembled chromatograms and confirmed sequence quality using Geneious v. 7.1.7 (Biomatters, San Francisco, California, USA). Sequences were aligned using ClustalW (Larkin et al., 2007) with default parameters using the Geneious platform. The alignment was confirmed by eye and translated into amino acid sequences to verify that no pseudogenes were present. Sequences were deposited in GenBank (KP898760–KP898867)

Nucleotide diversity, haplotype diversity, Tajima’s D and Fu’s Fs were calculated using DnaSP (Librado & Rozas, 2009). An Analysis of Molecular Variance (AMOVA) was performed using Arlequin (Excoffier & Lischer, 2010). To examine relationships between haplotypes, we conducted a Bayesian analysis using Mr.Bayes, accessed through Geneious. The best-fit model for the Bayesian analysis was selected using the corrected Akaike Information Criterion (AICc) with jModeltest 2.1.6 (Guindon & Gascuel, 2003; Darriba et al., 2012), based on 3 substitution schemes for compatibility with Geneious. The settings in Mr. Bayes were Nst = 6, rates= invgamma, ngammacat = 4,1,100,000 generations, sampling frequency = 1,000, number of chains = 4; temperature = 0.2, and burn-in = 100.

We examined the geographic distribution of the major well-supported haplogroups recovered in the Bayesian analysis. A Mantel test was conducted using the Isolation By Distance Web Service v. 3.23 (Jensen, Bohonak & Kelley, 2005) to test for isolation by distance. Pairwise geographic distances were calculated using Google Earth following the coast with the segments connecting two shoreline points ≤20 km, reflecting plausible larval transport routes and dispersal distances. We also compared intraspecific divergences between sequences from the major haplogroups with C. proteus, a cryptic sibling species of C. fragilis (Genbank accession numbers FJ858021–FJ858040, Wares, 2001).

Results

After trimming the ends and removing 6 positions with ambiguous base calls, our alignment was 613 base pairs, with 93 unique sequences (haplotypes), and 110 polymorphic sites, of which 58 were parsimony informative. In the amino acid alignment (which included the 6 positions excluded in the nucleotide alignment), there were three amino acid substitutions: a valine for an alanine in position 6 in a Charleston, South Carolina sequence; a valine for an isoleucine in position 55 for a Woods Hole, Massachusetts sequence, and an alanine for a threonine in position 157 for a Summerland Key, Florida sequence.

For all sites, haplotype diversity was high and Tajima’s D and Fu’s Fs were negative (Table 1), which may indicate population expansion or purifying selection. However, there were no trends with latitude and none of the Tajima’s D values were significant. FST and AMOVA results showed significant genetic structure particularly between distant sites (Table 2), with ∼14% of the variation among populations and ∼86% of the variation within populations (Table 3). The best-fit model selected using the AICc was HKY + I + G. A Bayesian analysis conducted with this model revealed three distinct, well-supported haplogroups (i.e., clades) (Fig. 1). A neighbor-joining tree based on HKY distances also uncovered these three haplogroups (Fig. 1), and was used to assess the distinctiveness of the haplogroups with the Species Delimitation Plugin in Geneious (Rosenberg, 2007; Masters, Fan & Ross, 2011). Within each of the three haplogroups, intraclade distances were significantly smaller than interclade distances (Table 4). Rosenberg’s PAB was 6.5E-18, 6.5E-18, 8.0E-34, for clades 1, 2, and 3, respectively (Table 4), strongly supporting reciprocal monophyly of the three haplogroups. All three haplogroups are clearly differentiated from the sister taxon Chthamalus proteus (Fig. 2).

Figure 1 Neighbor-joining topology generated using HKY distances.

The three major haplogroups are highlighted and the posterior probabilities obtained in the Bayesian analysis for the nodes defining these clades are given. SAND, Sandwich, Massachusetts; TR, Truro, Massachusetts; WH, Woods Hole, Massachusetts; RWU, Bristol, Rhode Island; VA, Gloucester Point, Virginia; SC, Charleston, South Carolina; SI, Sapelo Island, Georgia; SA, Savannah, Georgia; SK, Summerland Key, Florida; KL, Key Largo, Florida; TF, Tampa, Florida.

Figure 2 Comparison with Chthamalus proteus.

Midpoint-rooted neighbor-joining topology generated using HKY distances.

Table 2 Population paiwise FST values.

Distance method: pairwise distance. Negative values converted to 0. Significant values (p < 0.05) in bold.

	WH	RI	SC	TF	SI	VA	TR	SN	KL	SK	
WH											
RI	0										
SC	0.153	0.299									
TF	0.349	0.497	0.118								
SI	0.154	0.310	0	0.055							
VA	0.085	0.186	0.003	0.109	0						
TR	0.031	0.053	0.078	0.240	0.062	0					
SAND	0	0	0.202	0.381	0.205	0.103	0				
KL	0.100	0.241	0	0.120	0	0.018	0.080	0.154			
SK	0.158	0.306	0	0.044	0	0	0.074	0.203	0		
SA	0.390	0.537	0.103	0.112	0.023	0.137	0.267	0.430	0.143	0.065	

Table 3 Analysis of Molecular Variance (AMOVA) results.

Source of variation	d.f.	Sum of squares	Variance components	Percentage of variation	
Among populations	10	121.666	0.75549 Va	13.61	
Within populations	97	464.982	4.79363 Vb	86.39	
Total	107	586.648	5.54912		
Fixation index	FST: 0.13615				

Table 4 Species delimitation results.

Clade support is posterior probability from the Bayesian analysis for the node defining the clade (Fig. 1).

	Clade 1	Clade 2	Clade 3	
Closest clade	2	1	2	
Intra Dist	0.008	0.009	0.010	
Inter Dist—closest	0.020	0.020	0.024	
Intra/Inter	0.39	0.44	0.40	
P ID (Strict)	0.87 (0.82, 0.93)	0.86 (0.81, 0.91)	0.87 (0.82, 0.92)	
P ID (Liberal)	0.96 (0.94, 0.99)	0.96 (0.93, 0.99)	0.96 (0.94, 0.99)	
Av(MRCA-tips)	0.0057	0.0050	0.0086	
P(randomly distinct)	1.00	0.51	1.00	
Clade support	0.94	0.98	1	
Rosenberg’s P(AB)	6.5E-18	6.5E-18	8.0E-34	

Table 5 Distribution of individuals in haplogroups.

Number of individuals falling in haplogroups 1, 2, and 3 for each sampling location.

Sampling location	Clade 1	Clade 2	Clade 3	
SAND	4	6	1	
TR	1	5	3	
WH	6	4	2	
RWU	4	7	0	
VA	2	3	6	
SC	4	0	8	
SI	2	0	5	
SA	0	0	7	
SK	2	0	5	
KL	6	0	8	
TF	0	0	7	
Notes.

SAND Sandwich, Massachusetts

TR Truro, Massachusetts

WH Woods Hole, Massachusetts

RWU Bristol, Rhode Island

VA Gloucester Point, Virginia

SC Charleston, South Carolina

SI Sapelo Island, Georgia

SA Savannah, Georgia

SK Summerland Key, Florida

KL Key Largo, Florida

TF Tampa, Florida.

Haplogroups differed in their geographic distribution (Table 5; Fig. 3). Haplogroup 1 was present in all New England sites and most southern sites, except Savannah and Tampa. Haplogroup 2 was well-represented in the Massachusetts and Rhode Island sites, and also present in Virginia, but not in any of the more southern sites. Haplogroup 3 was present in the Sandwich, Truro, and Woods Hole, Massachusetts sites, but not in Rhode Island. It was the most abundant haplogroup in all of the southern sites. In Savannah and Tampa, it was the only haplogroup found. The Mantel test indicated significant isolation by distance (p < 0.001).

Figure 3 Geographic distribution of the three major haplogroups.

Discussion

Lineage diversity

Our results indicate significant genetic structure, with a break occurring between Virginia and South Carolina. We recovered 3 well-supported, reciprocally monophyletic COI haplogroups. One lineage was found in all locations, one in most locations (except Tampa and Savannah), and one in Virginia and northward locations only. Additionally, we observed significant genetic structure between northern and southern populations. This pattern—a cline between divergent clades—is similar to that observed for other barnacles, including Balanus glandula along the California coast (Sotka et al., 2004), Notochthamalus scabrosus along the Chilean coast (Zakas et al., 2009) and Chthamalus moro in southeastern Asia (Wu et al., 2014).

A deep phylogeographic break for species like barnacles with high planktonic dispersal potential may be due to several non mutually exclusive factors, including selection, cryptic speciation, and the presence of dispersal barriers (Zakas et al., 2009). It is possible that C. fragilis belonging to haplotype group 2 have characteristics that are less suited to southern locations. Additional research on the physiology and ecology of C. fragilis are necessary to elucidate possible adaptive differences between northern and southern populations.

The pattern of reciprocal monophyly and large between-clade relative to within-clade divergences can sometimes be used to infer the existence of cryptic species (Govindarajan, Halanych & Cunningham, 2005). Mitochondrial COI is used as a marker in many population-level studies, and as a genetic barcode to discriminate species (Bucklin, Steinke & Blanco-Bercial, 2011). While evolutionary rates differ between lineages, sequences originating from different individuals within a species show less divergence (often less than 3%, Bucklin, Steinke & Blanco-Bercial, 2011) than sequences originating from individuals belonging to different species (often >10%, Bucklin, Steinke & Blanco-Bercial, 2011).

Cryptic speciation may be common among chthamalid barnacles. Dando & Southward (1980) identified Chthamalus proteus as a cryptic species distinguishable only through molecular techniques from C. fragilis using enzyme electrophoresis, and these results were supported by Wares (2001) and Wares et al. (2009) using DNA sequences. In the Asian Chthamalus moro, Wu et al. (2014) observed interpopulation COI variation 3.9–8.3%, and inferred a cryptic speciation noting that population comparisons at the upper end of that range were comparable to interspecific divergence in the chthamalids Euraphia rhizophorae and E. eastropacensis (∼9%; Wares, 2001), which were separated by the rise of the Panamanian isthmus. However, the relatively short distances between our three C. fragilis clades relative to C. proteus do not support separate species status for the clades.

Our observed phylogeographic transition between Virginia and South Carolina spans Cape Hatteras, a region thought to be an important biogeographic boundary. Pappalardo et al. (2014) found that Cape Hatteras is a northern boundary for many species, but less so a southern boundary. In our dataset, this region is apparently a southern boundary for haplogroup 2. However, additional fine scale sampling between Virginia and South Carolina, especially around Cape Hatteras, is necessary to demarcate the location and nature of the break (e.g., Jennings et al., 2009).

Though a statistically significant pattern of isolation by distance (IBD) is detected in our data, we are cautious about interpretation. The strict interpretation of IBD is an equilibrium pattern between genetic drift and gene flow when migration is limiting, and so allele frequencies become divergent over spatial distance. However, similar statistical patterns emerge by non-trivial disjunct distributions of divergent lineages (Wares & Cunningham, 2005; Moyle, 2006), and may be driven by mechanisms of vicariance and selection on these divergent lineages. Given the high potential for larval dispersal in C. fragilis, we simply note that this statistical signal indicates a limit to gene flow, which may or may not be distinct from patterns of larval dispersal.

Northern expansion

Anthropogenic factors influence species distributions and population structure, which may facilitate the northward expansion of C. fragilis (Carlton, Newman & Pitombo, 2011). For barnacles, larvae can be transported long distances in ballast water and adults on ship hulls (Godwin, 2003; Zardus & Hadfield, 2005; Carlton, Newman & Pitombo, 2011). Coastal development is creating more and novel habitats for barnacles as well as other hard substrate organisms in regions dominated by sandy and muddy habitats where suitable substrate may have been previously limiting (Landschoff et al., 2013). Furthermore, warmer temperatures associated with climate change are thought to facilite poleward range expansions for many species (Barry et al., 1995; Zacherl, Gaines & Lonhart, 2003; Perry et al., 2005; Sunday, Bates & Dulvy, 2012), including barnacles (Southward, 1991; Dawson et al., 2010; De Rivera et al., 2011).

Here, we sought to gain insight into the origin of the northern expansion of C. fragilis. Carlton, Newman & Pitombo (2011) speculated that C. fragilis may have colonized Massachusetts by traveling on ships bound for Woods Hole from South Carolina. Alternatively, non-transport related anthropogenic factors may have facilitated expansion from the historical northern boundary in the mid-Atlantic. Warmer temperatures may have shifted ecological interactions to favor C. fragilis (Wethey, 2002; Carlton, Newman & Pitombo, 2011). Additionally, coastal development could have facilitated stepwise northward dispersal. Construction of marinas, docks, jetties, seawalls, and other structures provided hard substrate habitat that was not previously available in the typically sandy coastline between Chesapeake Bay and New England.

The absence of clade 2 south of Virginia suggests that northern C. fragilis likely originate from the northern part of its mid-19th century range. While our sample sizes are relatively small and we analyze a single marker, the complete absence of any haplogroup 2 sequences south of Virginia supports this hypothesis. Additional sampling in the mid-Atlantic region and analysis of multiple genetic markers will be crucial for both providing additional testing of this hypothesis, and for understanding the nature of the putative Cape Hatteras biogeographic break for C. fragilis.

As temperatures continue to increase, C. fragilis will likely continue to expand northward. Like C. fragilis, S. balanoides appears to be shifting its range poleward; however the mechanism driving the shift in S. balanoides is a range contraction in the southern part of its range (Jones, Southward & Wethey, 2012). Likely the range contraction is due to thermal stress in this boreo-arctic species, rather than interaction with encroaching C. fragilis. Further research is needed to understanding the potential impacts of range shifts on community dynamics (Sorte, Williams & Carlton, 2010). Our genetic analysis of C. fragilis, while limited, suggests that shifts in geographic distribution may be accompanied by shifts in genetic composition (e.g., expansion of haplogroup 2). Understanding how the population genetic composition is shifting, and how these changes may impact the overall community structure, is critical for understanding the consequence of climate change on coastal communities.

We thank V Starczak (WHOI) for assistance with collecting barnacles, P Polloni (WHOI) for assistance with laboratory procedures, and T Shank (WHOI) for use of his laboratory equipment.

Additional Information and Declarations

Competing Interests

Author Contributions

DNA Deposition

The authors declare there are no competing interests.

Annette F. Govindarajan conceived and designed the experiments, performed the experiments, analyzed the data, contributed reagents/materials/analysis tools, wrote the paper, prepared figures and/or tables, reviewed drafts of the paper.

Filip Bukša and Katherine Bockrath performed the experiments, analyzed the data, reviewed drafts of the paper.

John P. Wares conceived and designed the experiments, performed the experiments, analyzed the data, contributed reagents/materials/analysis tools, reviewed drafts of the paper.

Jesús Pineda conceived and designed the experiments, performed the experiments, analyzed the data, contributed reagents/materials/analysis tools, prepared figures and/or tables, reviewed drafts of the paper.

The following information was supplied regarding the deposition of DNA sequences:

Sequences were deposited in GenBank. Accession numbers KP898760–KP898867.

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
