# Peer review of "Phylogeographic structure and northward range expansion in the barnacle Chthamalus fragilis"

_PeerJ, doi:10.7717/peerj.926_

## Round 0.1 · accepted · Accept

It is my rare pleasure to report back that a paper can be accepted without revision, but both referees were in agreement that the paper is so well done that it can move forward as it is. I look forward to seeing this online.

Reviewer 1 ·

Basic reporting

No comments. Meets standards.

Experimental design

It would be nice to see more sampling along the gradient in the Mid-Atlantic region. This could be done in conjunction with possibly detecting the qualitative connection between South Carolina and Woods Hole stated in Carlton et al. (2011).
Although the clear break and structure shown with the single marker used in this study is enough support this particular manuscript to stand on its own.

Validity of the findings

No comments.

Additional comments

There are many qualitative statements on anthropogenic influence through habitat and transport. Chthamalus has been shown to take advantage of niches on vessel hulls and show flexibility in natural and man-made shoreline habitats. This provides qualitative support for these statements and therefore makes them valid for this manuscript.

·

Basic reporting

I enjoyed reading this study on the phylogeography of C. fragile along the US Atlantic coast. While there is only so much one can say given that the authors only used one marker and the geographic sampling was rather sparse, I feel like the analysis is publication worthy given the relatively strong results. The study opens up the possibility for a more in depth study that integrates molecular data, ecology, history, habitat availability, and potentially, the role of predator prey interactions, all of which I found to be important factors for explaining a range expansion along the California/Baja coast (e.g. Fenberg et al. 2014 - “Historical and recent processes shaping the geographic range of a rocky intertidal gastropod: phylogeography, ecology, and habitat availability”).

Experimental design

They followed an appropriate design for a standard CO1 phylogeography study.

Validity of the findings

I agree with their findings of the paper, which is suggestive of a range expansion of this species from the Virginia area towards New England over the past 100 years or so.

Additional comments

I look forward to seeing this paper in print soon.